

# Exploring the attitudes of medical faculty members and students in Pakistan towards plagiarism: a cross sectional survey

Farooq Azam Rathore[1], Ahmed Waqas[2], Ahmad Marjan Zia[2], Martina Mavrinac[3] and Fareeha Farooq[4]

[1] Department of Rehabilitation Medicine, Combined Military Hospital, Lahore Cantt, Pakistan
[2] CMH Lahore Medical College and Dental College, Lahore Cantt, Pakistan
[3] Department for Medical Informatics, University of Rijeka, School of Medicine, Rijeka, Croatia
[4] Department of Biochemistry, Fatima Memorial Hospital Medical College, Shadman, Lahore, Pakistan

Corresponding author
Ahmed Waqas,
ahmedwaqas1990@hotmail.com

## ABSTRACT

**Objective.** The objective of this survey was to explore the attitudes towards plagiarism of faculty members and medical students in Pakistan.

**Methods.** The Attitudes Toward Plagiarism questionnaire (ATP) was modified and distributed among 550 medical students and 130 faculty members in 7 medical colleges of Lahore and Rawalpindi. Data was entered in the SPSS v.20 and descriptive statistics were analyzed. The questionnaire was validated by principal axis factoring analysis.

**Results.** Response rate was 93% and 73%, respectively. Principal axis factoring analysis confirmed one factor structure of ATP in the present sample. It had an acceptable Cronbach's alpha value of 0.73. There were 421 medical students (218 (52%) female, 46% 3rd year MBBS students, mean age of 20.93 ±1.4 years) and 95 faculty members (54.7% female, mean age 34.5 ±8.9 years). One fifth of the students (19.7%) trained in medical writing (19.7%), research ethics (25.2%) or were currently involved in medical writing (17.6%). Most of the faculty members were demonstrators (66) or assistant professors (20) with work experience between 1 and 10 years. Most of them had trained in medical writing (68), research ethics (64) and were currently involved in medical writing (64). Medical students and faculty members had a mean score of 43.21 (7.1) and 48.4 (5.9) respectively on ATP. Most of the respondents did not consider that they worked in a plagiarism free environment and reported that self-plagiarism should not be punishable in the same way as plagiarism. Opinion regarding leniency in punishment of younger researchers who were just learning medical writing was divided.

**Conclusions.** The general attitudes of Pakistani medical faculty members and medical students as assessed by ATP were positive. We propose training in medical writing and research ethics as part of the under and post graduate medical curriculum.

## INTRODUCTION

Plagiarism is defined as "the deliberate or reckless use of someone else's thoughts, words or ideas as one's own, without clear attribution of their source" (*Mason, 2009*). It is a serious offense in academia and a major ethical concern which has received a lot of global attention in biomedical writing. There has been an increase in the number of manuscripts published on plagiarism in the last one decade. However most of the work is from the developed nations of the world where research training is usually imparted at undergraduate level. In comparison, the research output from developing countries including Pakistan is low, so there is a need to promote research education and training in these regions.

Plagiarism has been documented and reported mainly from the developed countries with a better research environment, stronger training and more common use of plagiarism detection software. With advancement in plagiarism detection software, an ever-increasing number of plagiarized papers are being recognized, often leading to their retraction from the journals. Employing plagiarism detection software and manual verification, *Baždarić et al. (2012)* reported the prevalence of plagiarized manuscripts among manuscripts submitted to Croatian medical journal during 2009–2010 to be 11% (85/754). A recent study of 2,047 cases of retracted papers from PubMed indexed journals reported an encouraging trend in recognition and retraction of plagiarized articles (*Steen, Casadevall & Fang, 2013*). While these statistics are encouraging, but most of the time detection after publication cannot prevent the damage that had already been done to science if plagiarized articles had already been cited. Retraction watch (http://retractionwatch.com/) is a blog which documents plagiarism, fabrication and retractions in the scientific community. It mentions a high number of research articles based on fake data, image manipulation, self-plagiarism, fake peer reviews and disputed authorships that are being retracted frequently from reputable journals (*Marcus & Oransky, 2014*). Unfortunately, this misconduct not only involves novice researchers, doctorate and post doc scholars from middle income countries but also scientists and institutes from Europe, Americas and Japan enjoying international fame and prestige. Plagiarizing research work often leads to great setbacks in one's careers.

Although the prevalence of intentional plagiarism in low resource countries has not been reported, it can be argued that it might be more prevalent in countries like Pakistan due to "a general lack of information regarding plagiarism among medical students and faculty members" (*Shirazi, Jafarey & Moazam, 2010*). However, the probability of intentional plagiarism both in the faculty and students also cannot be ignored. Prevalence of plagiarism is very hard to measure but the investigations of attitudes can also give us an insight in this phenomenon. This opinion is reinforced by Ajzen's theory of planned behavior which assumes that human beings are rational: A preceding intention entailing attitudes, subjective norms and perceived behavioral control, is necessary to perform a specific behavior (*Ajzen, 1991*).

A number of studies conducted in Romania (*Badea-Voiculescu, 2013*), Pakistan (*Shirazi, Jafarey & Moazam, 2010*), Croatia (*Mavrinac et al., 2010*), Norway (*Hofmann, Myhr &*

*Holm, 2013*) and Iran (*Ghajarzadeh et al., 2012*; *Ghajarzadeh et al., 2013*) have reported a high prevalence of positive attitude among both students and faculty members towards plagiarism.

Ghias' survey on academic dishonesty in Pakistani medical students reported a high prevalence of medical students who were involved in copying verbatim from online or published sources, senior peers, class mates with or without their consent, fabricating data to show desirable results, forging professors' signatures, faking health certificates to justify absence and other such behaviors (*Ghias et al., 2014*). *Poorolajal et al. (2012)* reported an overall prevalence of plagiarism as 38% in an Iranian University. This trend decreased by 13% with one unit increase in knowledge of plagiarism. Similarly in India, a high prevalence of plagiarism was attributed to pressure to publish and lack of facilities and funding in private institutions (*Singh & Guram, 2014*). This calls for serious educational reforms and implementation of strict policies regarding plagiarism not only in university curriculum but also in lower grades.

Although many studies specifically on plagiarism have been published abroad, in Pakistan research on this specific subject is lacking. This study was conducted on a relatively large sample of medical students and faculty in seven private and public medical colleges. The Attitudes Towards Plagiarism questionnaire (ATP) was chosen to explore knowledge and attitudes of faculty members and medical students towards plagiarism. The original ATP had 29 items assessing positive, negative and subjective attitudes towards plagiarism (*Mavrinac et al., 2010*). This questionnaire was based on Ajzen's theory of planned behavior and has been validated in Croatia (*Mavrinac et al., 2010*). Subsequently it was extensively used in other countries for example, India (*Gomez, Nagesh & Sujatha, 2014*), Iran (*Ghajarzadeh et al., 2012*) and Romania (*Badea-Voiculescu, 2013*).

The current study was designed with two aims: (1) To explore the attitudes of Pakistani medical students and faculty towards plagiarism; (2) To study the association between formal training in research ethics, medical writing and attitudes towards plagiarism.

## MATERIAL AND METHODS

A cross sectional survey was designed and conducted in three private and four public medical colleges in Lahore and Rawalpindi (August 2013–January 2014). Permission was obtained from the Institutional review board of CMH Lahore Medical College.

### Questionnaire

To collect data, we used a questionnaire divided in of three sections. The first section documented demographics. The second had questions on participants' interest and formal training in research methodology, research ethics and involvement in medical writing. The third section consisted of modified version of ATP (*Mavrinac et al., 2010*). The questionnaire was used with permission and modified for our study population. It was not translated from the original English version, as English is the language of instruction in all medical schools in Pakistan.

## Pilot survey

A pilot survey was conducted and feedback received from faculty and students during the pilot survey resulted in removal of 4 items in order to adopt the scale to Pakistani culture and academic environment. It was further modified from five-point to a three-point Likert type scale (agree (coded as 3), neutral (coded as 2) and disagree (coded as 1)) to facilitate the responses. Factorial analysis was performed to confirm the factor structure of the modified questionnaire.

## Participants

Convenience sampling technique was employed. Sample size was calculated at 95% confidence level and 5% confidence interval. Questionnaires were distributed among 550 medical students and 130 faculty members in 07 public and private medical colleges of Lahore and Rawalpindi. All participants read and signed informed consent forms, which were returned with each completed questionnaire. Forms were personally distributed and collected by two of the authors (AM, AW). Response rate was 93.45% and 73.05% for medical students and faculty members respectively. Ninety three forms were discarded (Incomplete or missing data, duplicate entries etc.).

## Data analysis

Data was analyzed by SPSS v 20. To confirm the factor structure of the questionnaire principal axis factoring analysis was used. The reliability of the questionnaire was calculated using Cronbach's alpha.

Descriptive and inferential statistical test were employed to analyze the data. An independent sample $T$-test was run to analyze associations between formal training in research ethics, medical writing (yes/no) and scores on ATP (continuous variables). Chi Square goodness-of-fit statistics was run to analyze association between score ranges of ATP and respondent group (faculty/student). One way ANOVA was run between scores on ATP and job designation, experience and education level of faculty members.

# RESULTS

## Characteristics of respondents

Characteristics of respondents and their training in medical writing are given in Table 1. There were 421 medical students and 95 faculty members. Most of the students were females 218 (51.8%) and 3rd year MBBS students 192 (45.6%).

## Questionnaire validation

Principal axis factor analysis was run to confirm the factor structure of the Pakistan version of the ATP. However, unlike Croatian version of ATP, the three factor structure was not confirmed. In present study, one factor structure was determined by the Scree-test (Fig. 1), interpretability criteria and the reliability of the factor calculated with Cronbach's alpha ($\alpha = 0.73$). The obtained factor represents an overall attitude towards plagiarism consisting of positive attitudes, negative attitude and subjective norms. Table 2 presents the factor structure of the Attitudes Towards Plagiarism questionnaire with factor loadings.

**Table 1 Demographic characteristics of medical students and faculty members ($n = 516$).**

| Variables | | Medical students | Faculty members |
|---|---|---|---|
| Gender | Male | 203 (48.2%) | 43 (45.3%) |
| | Female | 218 (51.8%) | 52 (54.7%) |
| Designation | Demonstrator | – | 66 (69.4%) |
| | Assistant Professor | – | 20 (21.1%) |
| | Associate Professor | – | 6 (6.3%) |
| | Professor | – | 3 (3.2%) |
| Education | MBBS/MD | – | 46 (48.4%) |
| | Masters Degree | – | 26 (27.4%) |
| | Fellowship | – | 19 (20%) |
| Education from abroad | | – | 15 (15.8%) |
| Median age (min–max) | | 21 (17–28) | 32 (23–61) |
| Training in medical writing | | 83 (19.7%) | 68 (71.6%) |
| Training in research ethics | | 106 (25.2%) | 64 (67.4%) |
| Currently writing an article | | 74 (17.6%) | 64 (67.4%) |
| Mean score and SD on ATP | | 43.21 (7.1) | 48.4 (5.9) |

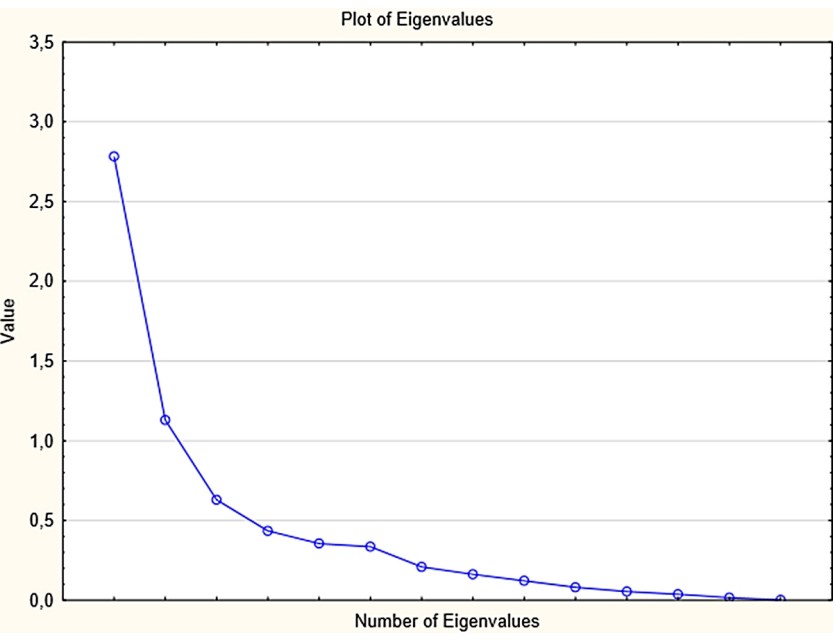

**Figure 1 Scree plot for the obtained one factor structure.**

Items 11, 12, 13 are not included in the final factor structure and analyses because of low factor loading ($<0.10$). The one factor structure explained 10.93% of variance in the questionnaire and average Inter-Item Correlation was .112. Total scores were obtained by summing all the statements. The mean score was divided into 3 ranges by 2 cut offs at 33.33%, 66.66% of scores on the ATP. Thus, scores on modified ATP were divided into

**Table 2 Factor structure of the attitudes towards plagiarism questionnaire with factor loadings.**

| Statements | Factor loading |
|---|---|
| 1. Since plagiarism is taking other people's words rather than tangible assets; it should NOT be considered as a serious offence. | −0.39 |
| 2. It is justified to use previous descriptions of a method, because the method itself remains the same. | −0.23 |
| 3. Self-plagiarism is not punishable because it is not harmful (one cannot steal from oneself). | −0.39 |
| 4. Plagiarized parts of a paper may be ignored if the paper is of great scientific value. | −0.36 |
| 5. Self-plagiarism should not be punishable in the same way as plagiarism is. | −0.13 |
| 6. Young researchers who are just learning the ropes should receive milder punishment for plagiarism. | −0.19 |
| 7. I could not write a scientific paper without plagiarizing. | −0.40 |
| 8. Short deadlines give me the right to plagiarize a bit. | −0.48 |
| 9. It is justified to use one's own previously published work without providing citation in order to complete the current work. | −0.39 |
| 10. Authors say they do NOT plagiarize, when in fact they do. | −0.20 |
| 11.[a] *Plagiarists do not belong in the scientific community.* | *0.09* |
| 12.[a] *The names of the authors who plagiarize should be disclosed to the scientific community* | *0.01* |
| 13.[a] *In times of moral and ethical decline, it is important to discuss issues like plagiarism and self-plagiarism.* | *0.02* |
| 14. A plagiarized paper does no harm to science. | −0.37 |
| 15. Sometimes one cannot avoid using other people's words without citing the source, because there are only so many ways to describe something | −0.30 |
| 16. If a colleague of mine allows me to copy from her/his paper, I'm NOT doing anything bad, because I have his/her permission. | −0.31 |
| 17. Those who say they never plagiarized are lying. | −0.37 |
| 18. Sometimes I'm tempted to plagiarize, because everyone else is doing it (students, researchers, physicians). | −0.32 |
| 19. I keep plagiarizing because I haven't been caught yet | −0.23 |
| 20. I work (study) in a plagiarism-free environment. | −0.16 |
| 21. Plagiarism is not a big deal. | −0.52 |
| 22. Sometimes I copy a sentence or two just to become inspired for further writing. | −0.41 |
| 23. I don't feel guilty for copying verbatim a sentence or two from my previous papers. | −0.32 |
| 24. Plagiarism is justified if I currently have more important obligations or tasks to do. | −0.52 |
| 25. Sometimes, it is necessary to plagiarize. | −0.51 |

**Notes.**
[a] Items 11, 12, 13 are not included in the final factor structure, because of to low (<0,10) factor loading.

three categories; low (<42), moderate (43–47) and high (>48). According to this scale, increasing score represents a positive leaning towards plagiarism.

## Attitudes towards plagiarism

Independent sample $T$-test revealed that those medical students who had been formally trained in medical writing were associated with low scores on the ATP (Mean group 1 = 43.61 (SD = 6.92), mean group 2 = 41.58 (SD = 7.51), $P < .05$), whereas students

**Table 3** Frequency distribution of medical students and faculty members in score ranges of ATP.

| Respondent | Low (<42.0) | Moderate (43–47) | High (>48) | Total ($n$) | $\chi^2$ value ($P$-value) |
|---|---|---|---|---|---|
| Medical student | 190 (45.1%) | 112 (26.6%) | 119 (28.3%) | 421 (100%) | 26.5 ($P < .001$) |
| Faculty member | 14 (14.7%) | 32 (33.7%) | 49 (51.6%) | 95 (100%) | 19.3 ($P < .001$) |

who were trained in research ethics ($P = .936$) or were currently writing a research paper ($P = .674$) did not differ from their counterparts on ATP scores.

The frequency distribution of medical students and faculty members between score ranges is given in Table 3. According to Chi Square goodness-of-fit statistics, a statistically higher percentage of faculty members had ATP scores in moderate 32 (33.7%) or high category 49 (51.6%) than low category 14 (14.7%) ($P < .001$). A higher proportion of students had ATP scores in the low (45.1%) or moderate category (26.6%) than in the high category (28.3%) ($P < .001$).

Pearson Chi-Square revealed that faculty with foreign qualifications had better formal training in research ethics ($p < 0.05$). According to it, all of the faculty member who were educated abroad ($n = 15$) had received formal education in research ethics. More than half (61.2%) of faculty members educated in Pakistan had a formal education in research ethics.

No statistically significant difference was found between mean scores on ATP scores and job designation ($P = 0.734$), experience levels ($P = 0.208$) education level ($P = .068$). Independent sample $T$ test revealed no significant association between ATP scores and training in research ethics ($P = .87$), medical writing ($P = .17$) or current involvement in medical writing ($P = .99$).

Table S1 gives response percentage of faculty members and students on modified ATP.

## DISCUSSION

The study revealed that the majority of medical students (55%) and faculty members (82.7%) had moderate or high scores on the ATP. This represents their approval of plagiarism. This finding is in consonance with previous studies conducted in Pakistan.

Lower scores on ATP in medical students were associated with training in medical writing whereas ATP scores were not significantly associated with formal education in research ethics or current involvement in medical writing. Factor analysis revealed a one-factor structure representing attitude towards plagiarism with 22 statements and good reliability. This version of ATP is valid and reliable for use on Pakistani population.

*Shirazi, Jafarey & Moazam (2010)*, have attributed lack of training in research methodology and referencing techniques among Pakistani students and faculty rather than malice as a cause of plagiarism in most cases. Shashok has also noted that many cases of plagiarism are unintentional and arise from lack of knowledge of citation practices, pressure to increase publication output, and inability to write and communicate ideas in English (which may lead to copy-pasting to improve use of language in the manuscript) (K Shashok, pers. comm., 2011).

Formal training in research methodology, medical and publication ethics at the undergraduate level is generally lacking in Pakistan. Even the faculty members are not clear about the definition, types and implications of plagiarism and unethical practices in medical writing and research. The mandatory training workshops of the college and physicians and surgeons in Pakistan for the trainees and supervisors do not adequately address plagiarism and other unethical practices in medical research and writing.

Only about one quarter of students in our sample were formally trained in medical writing and research ethics. These findings are consistent with those of *Shirazi, Jafarey & Moazam (2010)*, who have attributed lack of knowledge of proper referencing and citing as a cause of plagiarism in medical students. In contrast to students, most of the faculty members had received formal training and education in research ethics and medical writing. This was probably due to involvement of the faculty in the continuing medical education, self-directed learning and the recent revision of faculty promotion rules by the Pakistan Medical and Dental Association (PMDC) which mandates the faculty members to write a certain number of articles for promotions. The medical students who had been trained in medical writing or were currently involved in medical writing had a low tendency towards plagiarism. In our study, year of study did not affect attitudes towards plagiarism in medical students. These findings favor our hypothesis that formal education of medical students would decrease the prevalence of plagiarism. However, the evidence of efficacy of educational interventions on attitude towards plagiarism is rather confusing. An online case study discussing plagiarism by adult learners found no significant association between the incidence of plagiarism and cheating and educational interventions on policies related to academic honesty (*Jocoy & DiBiase, 2006*). However, imparting information related to policies on academic honesty immediately before examinations lowered the incidence of cheating behaviors (*Kerkvliet & Sigmund, 1999*). Similarly, *Anderson et al. (2007)* reported no significant association between attending formal courses on research ethics and academic dishonesty. However, it should be noted that results of these studies were from developed countries, therefore, these results might not be applicable in Pakistan where cultural and academic environment is very different.

Most of our sample of faculty members (69.4%) were less experienced demonstrators in medical colleges who did not have postgraduate degrees or fellowships yet. This highlights the need for continuing medical education programs and for interventions on research ethics and medical writing. *Ghajarzadeh et al. (2012)* reported similar trends in ATP scores of Iranian faculty members. These arguments are in consonance with *Shirazi, Jafarey & Moazam (2010)*, where less than 30% of the faculty members had correct knowledge of copyright rules, referencing or use of quotation marks.

Most of the students disagreed with the statement that they worked in a plagiarism free environment. Such high "perceived" prevalence of plagiarism among medical students might be rooted in the learning styles of most Pakistani students who, unlike students at Western institutions, are more involved in rote and teacher-centered learning (*Introna et al., 2003*). Many students tend to copy verbatim from learning resources or others' work mainly due to insufficient language proficiency (K Vessal & F Habibzadeh, pers.

comm., 2007). This behavior can be discouraged by increasing the awareness and use of plagiarism detection software among both faculty and students. As confirmed in our study, a high percentage of students resort to cheating behavior because they haven't been caught yet. This trend was also explored by another study which reported a very low awareness about existence of plagiarism detection software in Pakistani university students (*Ramzan et al., 2012*).

It is essential to understand its etiology if one is to decrease the practice of plagiarism. In a recent study the majority of the respondents (both medical students and faculty) confessed to having plagiarized at least once in their life (*Shirazi, Jafarey & Moazam, 2010*). This supports our findings, where only 24.2% of the medical faculty and 20.4% of medical students agreed that they worked in a plagiarism free environment. The causes of this evil practice in Pakistani medical faculty are manifold. PMDC has laid down strict criteria of qualification, teaching experience and research experience for promotion in academic ranks. Promotion of an assistant professor to rank of associate professor and to professor requires at least 3 and 5 publications respectively in PMDC indexed journals (*Pakistan Medical & Dental Council, 2011*). In our study, a majority of the faculty members and students agreed that approaching deadlines gave them a right to plagiarize (Table S1). Thus, approaching deadlines (pressure to publish) and promotions in academia have led to a focus on quantity rather than quality of research products.

A majority of the medical students agreed that young researchers should receive milder punishment, but medical faculty had mixed opinions. A majority of the respondents in our survey agreed that they are tempted to plagiarize because everyone else is doing it. Therefore, in our opinion the identities of the plagiarists should be brought to light to set an example for the academic community and keep plagiarism in check.

Proper policies should be devised by the stakeholders and training modules on research and medical writing should be introduced in medical curriculum. The Higher Education Commission, Pakistan (HEC), Pakistan Medical and Dental Council, Pakistan (PMDC) and College of Physicians and Surgeons, Pakistan (CPSP) should facilitate the medical colleges and universities to train and establish ethics review committees and intuitional review boards. The medical colleges should invest in plagiarism detection software and make it available to their students, trainees and faculty members. Even if an institute doesn't subscribe to a paid plagiarism detection software, there are many free alternatives available. Workshops, seminars, invited lectures should be arranged specifically to address this issue. Dedicated modules on research methodology, analytical and referencing techniques should be integrated in undergraduate medical curriculum to further develop the research environment in Pakistan. This calls for a revision of undergraduate and postgraduate curriculum and faculty training with an emphasis in teaching the current best practices and ethics of medical research and writing.

## LIMITATIONS

The cross-sectional design of this study limits inferences about causality and temporality. Use of self-administered questionnaires may lead to information bias. The present study

is based on an adequate sample size but it was collected using convenience sampling approach. Therefore, its results cannot be generalized to the whole Pakistani population or medical students or faculty members.

## CONCLUSIONS

The general attitudes of Pakistani medical faculty members and medical students as assessed by ATP were approving towards plagiarism. There is a lack of training in biomedical ethics and good practices in medical writing. We propose training in medical writing and research ethics as part of the under- and post-graduate medical curriculum. Faculty should keep itself updated about the latest policies regarding plagiarism inside the country and abroad. Steps should be taken by PMDC, CPSP and HEC to raise awareness about this menace in Pakistan.

## ACKNOWLEDGEMENTS

The authors thank Sana Gulraiz, student at Services Institute of Medical Sciences, Lahore and Bilal Gujjar, student at Sheikh Khalifa Bin Zayed Al Nahyan Medical and Dental College, Lahore, for their help in collecting data for this project. The authors also thank Andrew J. Haig, MD, Professor of Physical Medicine and Rehabilitation at the University of Michigan, USA and Dr. Colleen O'Connell, MD, FRCPC, Research Chief, Physical Medicine & Rehabilitation at Stan Cassidy Centre for Rehabilitation, Canada, for improving the use of language in the manuscript.

### Funding

The authors declare that they had no funding source for this project.

### Competing Interests

The authors declare that there are no competing interests.

### Author Contributions

- Farooq Azam Rathore conceived and designed the experiments, analyzed the data, contributed reagents/materials/analysis tools, wrote the paper, prepared figures and/or tables, reviewed drafts of the paper.
- Ahmed Waqas conceived and designed the experiments, performed the experiments, analyzed the data, wrote the paper, prepared figures and/or tables, reviewed drafts of the paper.
- Ahmad Marjan Zia conceived and designed the experiments, performed the experiments, wrote the paper, prepared figures and/or tables, reviewed drafts of the paper.
- Martina Mavrinac performed the experiments, analyzed the data, contributed reagents/materials/analysis tools, wrote the paper, prepared figures and/or tables, reviewed drafts of the paper.

- Fareeha Farooq analyzed the data, contributed reagents/materials/analysis tools, wrote the paper, prepared figures and/or tables, reviewed drafts of the paper.

## Human Ethics

The following information was supplied relating to ethical approvals (i.e., approving body and any reference numbers):

Permission was obtained from the Institutional review board of CMH Lahore Medical College. Ethical review letter is provided as a supplementary file.

## Supplemental Information

Supplemental information for this article can be found online at http://dx.doi.org/10.7717/peerj.1031#supplemental-information.

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
