# Peer review of "Exploring the attitudes of medical faculty members and students in Pakistan towards plagiarism: a cross sectional survey"

_PeerJ, doi:10.7717/peerj.1031_

## Round 0.1 · original submission · Major Revisions

Please pay particular attention to the comments from Reviewer 1. In particular, you might like to seek statistical advice to respond to her comments on the analysis. The manuscript would also benefit from a thorough language edit to remove grammatical errors and improve clarity. I very much hope you will make these revisions and then resubmit to PeerJ.

·

Basic reporting

Introduction:
Begin with the definition of plagiarism and the problem of it, then explain the importance of attitude, mention Ajzen's theory, then explain previous investigations and state the aim
- the explanation of Retraction Watch could be shortened
- Studies that explore attitudes towards plagiarism or research integrity are not mentioned in detail to get a better insight of the problem
• in Pakistan: Ghias et al (Rennie's questionnaire): http://www.ncbi.nlm.nih.gov/pmc/articles/PMC4060764/
• in Iran: http://www.ncbi.nlm.nih.gov/pmc/articles/PMC3521886/
• in India: http://www.ncbi.nlm.nih.gov/pmc/articles/PMC3938877/
- Reference Marcus & Ivan, 2014 – Marcus&Oransky,2014
- Mavrinac et al are from Croatia, correct the sentence in this section
- as the whole investigation is based on Ajzen's theory of planned behaviour a brief explanation of connection between attitudes and subjective norms and behaviour has to be mentioned

Experimental design

Materials and Methods:

- the sample is convenient, not representative but big enough and from seven different institutions, public and private; the ratio female/male is almost 50%-50% which is ok
- please indicate the median age (minimum-maximum)
- it is not clear what a low positive attitude result means, you have to indicate how did you count the total points on the scale; what is low, medium and high attitude? How did you decide to call a result low or medium?
- „The Cronbach’s Alpha value of the modified version of ATPQ was 0.740 reflecting the good reliability of this scale.“ – this is an acceptable value but it is not acceptable to count this value for the whole questionnaire as there are three factors and each have to be counted independently.

Data analysis:
First of all, you have to validate the questionnaire on your sample using PCA factor analysis.Then you have to analyze the factors (you can have the same or different factors) , factor loadings, and % of explained variance, and afterwards you have to calculate the reliability of each factor. You have to calculate the correlations between the factors and present them in results.
After you have obtained your factors the analysis can be done. You can compare attitudes of your groups and attitudes according the independent variables.

- t-test can be used only to asses difference between groups on conutinuos variables, for categorical variables Chiu-square test has to be used
- Pearson Chi Square cannot be used analyze associations between categorical
variables, only to assess correlation of continuous variables
Results:
The tables should be:
1. Demographic data of participants
2. Validation of ATPQ – factors, items and factor loadings, reliability (Cronbach alpha)
3. Table one and two could be united in one table, put subscales in rows and subsamples in columns
4. Mean scores on subscales for the subgroups according to independent variables (mean values± SD and exact values if normal distribution or non-parametric statistics: median (5,95 percentile), differences tested with Mann-Whitney test or Kruskal-Wallis test)

- the demographic data would be much more clear if put in table 1, divided in two columns, one for students and another for stuff
- Independent sample T-test revealed that those medical students who had been formally trained in medical writing had less positive attitudes towards plagiarism (Mean difference= 1.43, P < .01). – indicate mean±SD for each group
- subjective attitudes are norms, r=.553, p value (two-tailed <.01). – put the exact p value in the text
- Pearson Chi-Square revealed that Faculty with foreign qualifications had better formal training in research ethics (p-<0.05). – This is not clear, where are the data that show this?
- Put the exact p value in brackets: No statistically significant difference was found between mean scores on positive attitudes (P=?), negative attitudes (P=?), subjective attitudes and job designation, training in research ethics and medical writing and current involvement in medical writing. (P value
>0.05)
- tables 3,4 and 5 are sufficient, could be put as additional material. There are sufficient because the questionnaire has 3 factors/subscales and there is no need to analyze each item although interesting.

Validity of the findings

Discussion:
- the structure of this section is not clear: Begin the discussion with your major finding, not the definition of plagiarism, do NOT repeat the introduction; Then comment your results as stated in the Results section and compare them with the literature and speculate.

Additional comments

Dear authors,

your article is very interesting and one of the few that deals with the investigation of plagiarism using a validated questionnaire and association with training. Although very interesting, it has problems with the statistical analysis (the new version of the ATPQ was not validated for this sample, the confirmation factor analysis has to be done in order to validate the factors; then the reliability (Cronbach alpha) has to be accesed for each factor
) and presentation of the results that have to be corrected if the manuscripts is going to be accepted for publishing. See the comments.

Reviewer 2 ·

Basic reporting

The article has the required structure. However, it would be benefit from a careful reading by the authors for grammatical corrections

Experimental design

Adequate

Validity of the findings

Adequate

·

Basic reporting

This is an important and interesting study.

Experimental design

Approriate and adequate for the stated objectives.

Validity of the findings

Valid and reliable.

Additional comments

An important contribution to the neglected area of academic plagiarism in developing countries.

---

## Round 0.2 · Major Revisions

The reviewers agree that you have made important improvements to your article but Dr Bazdaric has provided further, detailed suggestions which I think are helpful so I hope you will agree to revise the manuscript further.

I suggest you pay particular attention to ensuring that the conclusion reflects your findings and that the abstract is revised in line with the body of the text.

I note that the reviewer offered to supply a Word document rather than a PDF -- I have asked PeerJ if it is possible to supply this to you. Please let PeerJ know if you would like to receive the Word document and I will see if this can be facilitated.

·

Basic reporting

The article is now significantly improved. English has to be checked by a native speaker after all corrections.
See comments in word document.

Experimental design

The results are improved.

Validity of the findings

The authors have followed my comments and done the factor analysis as proposed. The results differ from the original version of the questionnaire, which is not unusual,-

Additional comments

Dear author, you have improved your manuscript significantly and I am really satisfied with the work.But I still have comments, and would like to see once again your manuscript.
Abstract: has to be improved, it was not corrected after the factor analysis.
Introduction: The aim is meaningful and logic, but the text in introduction has to lead to the aim (see my comments in text).
Results: correction of table 3; Table 3: Frequency distribution of medical students and faculty members in score ranges of ATPQ - there should ba a sum for columns and test statistics positioned in the last column.
Discussion: see my comments. Generaly, it has to follow the results section.

Reviewer 2 ·

Basic reporting

.

Experimental design

.

Validity of the findings

.

Additional comments

I have reviewed this paper previously and said it was acceptable. I see that the authors have revised the paper in response to another reviewers comments. I agree the paper reads better than before. In my opinion it is a useful piece of work that should see the light of day.

---

## Round 0.3 · accepted · Accept

Thank you for responding to the reviewers' detailed suggestions. We hope you agree that the resulting revised paper has been substantially improved. I believe it is now suitable for publication.